# Predictive Value of Sonographic Parameters on the Effects of Cranial Molding Helmet Therapy in Infants with Positional Plagiocephaly

**DOI:** 10.3390/diagnostics14131407

**Published:** 2024-07-01

**Authors:** Maria Licci, Agnes Paasche, Alexandru Szathmari, Pierre-Aurélien Beuriat, Carmine Mottolese, Raphael Guzman, Federico Di Rocco

**Affiliations:** 1Department of Pediatric Neurosurgery, French Referral Center for Craniosynostosis, Hôpital Femme Mère-Enfant, Hospices Civils de Lyon University of Lyon, INSERM 1033, 69500 Bron, Lyon, France; agnes.paasche@chu-amiens.fr (A.P.); alexandru.szathmari@chu-lyon.fr (A.S.); pierre-aurelien.beuriat@chu-lyon.fr (P.-A.B.); carmine.mottolese@chu-lyon.fr (C.M.); federico.dirocco@chu-lyon.fr (F.D.R.); 2Department of Neurosurgery, Division of Pediatric Neurosurgery, University Hospital and University Children’s Hospital of Basel, 4031 Basel, Switzerland; raphael.guzman@usb.ch

**Keywords:** positional plagiocephaly, non-synostotic plagiocephaly, ultrasonography, occipital angle, bone thickness, helmet therapy, helmet treatment efficacy

## Abstract

Positional plagiocephaly is a deformational cranial flattening frequently treated in pediatric neurosurgical practice. Positional maneuvers and orthotic helmet therapy are preferred therapeutic options for moderate-to-severe forms. Treatment response seems to be age-dependent. Nevertheless, predictive data are vague, and cost-efficiency might be a limiting factor for treatment. The purpose of this study was to investigate the early predictive value of sonographic parameters on the efficacy of orthotic helmet therapy through the assessment of changes in skull shape and correlation of the parameters with caliper cephalometry values and with age. A consecutive cohort of 49 patients < 10 months of age, undergoing orthotic helmet therapy for positional plagiocephaly, was recruited prospectively. The authors routinely assessed the patency of the lambdoid sutures by ultrasound and the following additional skull parameters were measured: suture width, adjacent full bone thickness, adjacent cortical bone thickness and occipital angle. Caliper cephalometric values, as well as demographic and clinical data were collected. Retrospective data analysis showed an inverse relation between both cortical and full skull bone thickness and early treatment efficacy, defined by a reduction in the occipital angle. The improvement of sonographic parameters correlated with the development of cranial caliper cephalometry values. In conclusion, the sonographic assessment of skull bone thickness is a safe and cost-effective tool to predict the early efficacy of orthotic helmet therapy in positional plagiocephaly and might, therefore, help the clinician to foresee the potential evolution of the deformity.

## 1. Introduction

Deformational posterior plagiocephaly, also referred to as occipital plagiocephaly, is defined as a non-synostotic, asymmetric deformity of the head as a result of external forces, acting on the pliable infant skull either in utero or postnatally, and persistent in infants after six weeks of age [1]. It is a relatively common entity that occurs in up to 1 of 5 infant and peaks in the first six months of life [2,3,4,5]. In opposition, the prevalence of the premature fusion of the lambdoid suture, which is also causing posterior plagiocephaly, but is associated with the growth restriction of the skull, and therefore, with a different head shape, is estimated at a rate of 3 in 100,000 infants and represents the rarest form of true single-suture synostosis. In addition, lambdoid synostosis is rarely isolated and most cases are found to be syndromic in a large spectrum of diseases [6]. The majority of cases of anterior plagiocephaly are caused not by positional factors but by unilateral coronal synostosis [7,8,9]. In addition to the evaluation of well-established clinical parameters, according to phenotypical features of the head shape, nowadays, the diagnostic differentiation between a positional deformity and a true synostosis can be easily and safely performed by cranial ultrasound, reducing the need for exposing infants to ionizing radiation [10,11,12,13,14,15]. The treatment options are controverse and outcome predictors are rare, although the case numbers are high. In fact, over the course of the past 30 years, the overall prevalence of positional posterior plagiocephaly has risen, resulting from the American Academy of Pediatrics 1992 recommendation to place babies in the supine sleep position in order to reduce the rate of sudden infant death—and from additional, pregnancy-related risk factors such as increased maternal age [4,5,9]. However, the point prevalence of positional plagiocephaly appears to be age-dependent and may be as high as 22.1% at 7 weeks of age, with a favorable tendency to decrease to a value as low as 3.3% at 2 years, consistent with the physiologic rate of head growth [9,16,17]. Considering the natural course and the interventional risks, most craniofacial surgeons therefore agree on the low value of surgery [7,18,19]. Other than the observation of the natural course, however, active therapeutic options include physiotherapy, osteopathy, counter-positioning maneuvers and external orthotic helmets, the latter currently having a leading role in cranial remodeling for moderate-to-severe forms [2,16,20,21,22,23,24]. These forms are likely to correct more quickly with cranial orthosis and the ideal time to start helmet therapy has been reported before 6 months of age [25,26]. In fact, multiple studies show an inverse correlation between younger age at the onset of orthotic remodeling and treatment efficacy, most likely related to the underlying pathogenetic mechanism of deformational deformity, consisting in a higher bone remodeling potential of the skull during the first months of life [2,25,27]. However, there are reports demonstrating that also children older than 12 months of age with deformational plagiocephaly may still benefit from orthotic therapy, and data on long-term outcomes are generally still debatable [23,26]. Overall, it is difficult to know at an early stage whether children with deformational plagiocephaly will undergo sufficient remodeling by counter positioning or if an orthotic helmet might be the most expedient procedure. Additional prognostic data regarding the effect of orthotic cranial remodeling are scarce and the treatment is expensive. Moreover, multiple helmet adjustments might be needed over time in order to cope with physiologic head growth, which is both cost-increasing, time-consuming and potentially associated with the side effects of a prolonged helmet therapy [28]. Therefore, the authors aimed at identifying positive predictive sonographic parameters on the early effect of cranial molding therapies. They hypothesize that the efficacy of orthotic cranial remodeling is related to the osseous characteristics of the infantile skull, with thinner cranial bones being more prone to mechanical remodeling. Herewith, infants that will truly profit from early treatment might be identified and costs could be deployed purposefully. 

## 2. Materials and Methods

We performed a retrospective analysis of a prospectively collected, consecutive pediatric cohort, including 49 infants of 4–10 months of age at the time of recruitment, receiving orthotic helmet therapy at our institution over a 6-month period from January to June 2021 for cranial molding in positional, posterior plagiocephaly. Infants older than 12 months at the onset of helmet therapy treatment were excluded, since treatment efficacy has been shown to be inversely related to age at the onset of treatment, and the goal of the study was to analyze early treatment efficacy. Cranial ultrasound was routinely used to assess the permeability of the suture, and therefore, exclude a true lambdoid synostosis, as part of the previously established diagnostic workup. Additionally, sonographic measurements of both skull and suture characteristics were performed at the onset of orthotic helmet treatment and at three-month follow-up time points (10–14 weeks). Clinical data assessment included the morphologic appearance according to the Argenta Classification (I–IV) of positional posterior plagiocephaly, the parental impression of the natural course, a potential adjuvant treatment at the time of recruitment (positional maneuvers, physiotherapy and/or osteopathy), the presence or absence of congenital torticollis, sleeping habits (average length of night sleep, sleeping position), the length of daytime ventral positioning, birth delivery modality (vaginal, vaginal assisted, C-section), as well as prematurity and twin or multiple pregnancy. Bird-eye-view pictures in standardized, lying, neutral position were taken, both at recruitment and at the three-month follow-up visit after the onset of treatment. Three-dimensional laser cranial reconstruction measurements were used to fabricate customized, active-molding orthotic helmets and to monitor the cranial remodeling over time, the latter in order to enable a digitalized comparison (Figure 1). Caliper cephalometry values were assessed at each time point. Cranial vault asymmetry (CVA mm), cranial vault asymmetry index (CVAI %) to assess plagiocephaly severity and Cephalic Ratio (CI %) to objectivize a potential, asymmetric brachycephaly component were calculated. Ear displacement was noted both according to clinical impression and to the 3D laser reconstruction model. Sonographic measurements were performed on an Esaote MyLab^TM^Seven Ultrasonograph using a 7.5–15 MHz multi frequency linear transducer (high resolution probe). Five ultrasonographic parameters were analyzed on bilateral lambdoid suture imaging (Figure 2): suture patency (yes/no); suture width (millimeters); adjacent full (diploic) bone thickness and upper cortical bone thickness (millimeters, within 5–10 mm from the suture center line on both sides on a perfect orthogonal projection); and occipital angle (angle between the straight lines drawn from each end point of the lambdoid suture along the skull). We applied the previously described 2-point measurement technique, foreseeing a vertical placement of the ultrasound probe on two trisected points of the suture line [29]. Suture width was defined in terms of a horizontal and vertical width as a mean of two measurements at the upper and lower end. Adjacent bone thickness was subdivided into upper cortical bone thickness and bi-cortical full bone thickness. Measurements were taken on both skull plates adjacent to the suture and the mean value was defined. Finally, the occipital angle was measured between the adjacent parietal and occipital bone on both the affected and unaffected skull, between the lines projected along the lamboid sutures of the skull, according to the technique previously described by Kim et al. [30]. The occipital angle ratio (OAR) was calculated as the mean occipital angle of the affected side divided by the angle of the unaffected side. All measurements were performed by a single senior investigator, routinely performing cranial ultrasound exams (ML). The infants were either placed in a prone or sitting position, depending on their age and preference, or they were held by their primary caregivers in an upright position. Through playing distraction or feeding, the ultrasound examination could be performed without discomfort and with a duration up to a maximum of five minutes, using the 2-point measurement method on both lambdoid sutures [29]. Data analysis was performed with IBM SPSS Software (SPSS Inc., Chicago, IL, USA; Version 25). Correlative exploration between sonographic parameters was performed using linear regression analysis, and ANOVA was applied for the intergroup correlation of clinical and radiological data. An interclass correlation coefficient was used to examine the intra–rater reliability of repeated occipital angle measurements.

## 3. Results

In total, 49 infants with positional posterior plagiocephaly were consecutively recruited in the study population over a six-month time period and follow-up measurements were taken at the onset of treatment and at three-month follow-up. Assumption checks confirmed a multivariate normality of the consecutive cohort. The relevant descriptive characteristics of the study cohort are summarized in Table 1. The cohort shows a male predominance (69.3%), according to the data available in the literature for infants with positional posterior plagiocephaly [5,31]. Treatment onset was under 6 months of age for most of the group (mean value 5.9 months, SD 1.2 months) and the infants that were included presented mostly with a moderate (44.9% Type 3)-to-severe (32.6% Type 4–5) Argenta grading. The vast majority presented with torticollis at birth as an underlying etiological risk factor (61.2%), had already undergone combined physiotherapy and osteopathy (63.2%), and the primary caregivers reported a steady state (57.1%) or progression (28.6%) of the deformity prior to the onset of helmet therapy. No difference in the treatment outcome was observed for gender or other pathology-related risk factors. The caregiver’s impression of the early therapeutic course after three months of orthotic cranial remodeling clearly shifted towards a reduction in the deformity (66.7%).

Ultrasonographic bone thickness analysis in relation to age at the onset of orthotic helmet treatment showed a higher variability for cortical bone thickness measurements than for full bi-cortical bone thickness (Figure 3). No linear correlation between either upper cortical bone thickness or full bi-cortical bone thickness and age was proved (*p* = 0.68 and *p* = 0.84) in our cohort.

Over a three-month follow-up period, the overall improvement of the deformity was objectively confirmed, with a reduction in the Argenta grade by 1 or 2 points (32.6 and 36.7%, respectively) in the majority of the cases (Figure 4). The CVAI (%) and CVA (mm) confirmed the findings and both the effective time of continuous treatment and the duration per day significantly correlated with the CVAI (%) reduction (*p* = 0.042 and *p* = 0.010, respectively). The effective time of treatment, however, differed from the follow-up time, with a realistic range of a continuous orthotic helmet treatment of 6–10 weeks instead of 10–14 weeks and a helmet treatment duration/day of 18 h (mean 18.33 h, SD 4.85 h).

Our results confirmed the positive correlation between the sonographically defined OAR and the CVAI (r = 0.293, *p* = 0.44). Furthermore, the objective radiologic evidence of the clinical amelioration was reflected by the ultrasonographic reduction in the occipital angle on the side of the deformity (Mean OAR reduction 8.3°, SD 5.8°, Variance 33.5°), as shown in Figure 5. Of the investigated, predictive ultrasonographical measurements, both the upper cortical bone thickness and the full bi-cortical bone thickness, showed an inverse relation with the improvement of the occipital angle (*p* = 0.033, 95% CI [0.03, 0.56] and *p* = 0.004, 95% CI [0.15, 0.64], respectively), as shown in Figure 6. Lower values of both cortical and full bone thickness correlate with an effective, early orthotic helmet remodeling. Although there was a tendency towards correlation, no significant predictive value was seen for the suture width (both for vertical and horizontal measurements). The intra–rater reliability of occipital angle measurements using ultrasound was high, accordingly to the results of previous studies. This result was in agreement with previous studies [15,30], investigating the patency of lambdoid sutures using ultrasound in infants with DP.

The sonographically assessed amelioration in head shape, defined by the OAR, was clinically reflected by the correlated improvement of cranial caliper cephalometric values (CVAI; *p* = 0.001, 95% CI [3.09, 7.09]), as shown in Figure 7. Overall, younger age was a determining factor for treatment success, defined as the sonographic improvement of the occipital angle (*p* = 0.009, 95% CI [0.64, 4.27]) and the clinical improvement of CVAI (*p* < 0.001, 95% CI [3.09, 7.09]). However, no clear correlation between age as a determining factor for treatment efficacy, and changes in the OAR in combination with bone thickness parameters could be determined, both for upper cortical and for full bone thickness assessment (*p* = 0.44 and *p* = 0.83, respectively).

## 4. Discussion

The prevalence of deformational posterior plagiocephaly has risen significantly after the recommendation to position healthy infants on the back or the side for sleep came up back in 1992 in order to lower the risk for sudden infant deaths syndrome (SIDS) [4,32,33]. In our cohort, indeed, most infants fully slept on the back, with only 16.3% being positioned on their belly during daytime sleep. Positional preference is the major cause of the deformity and additional risk factors include prematurity, male sex, maternal age (>35 years), twin or plurals pregnancy, assisted delivery, cephalohematoma and torticollis [16,25]. These risk factors were represented also in our cohort, with an emphasis on male sex (69.4%), torticollis (61.2%) and prematurity (10.2%). However, these factors did not influence early orthotic helmet treatment efficacy. In addition to distinctive clinical features that clearly help to differentiate positional plagiocephaly forms from true lambdoid synostosis in most cases, the use of ultrasonography has gained acceptance as a non-invasive tool to confirm the diagnosis, whereas the role of CT scans, widely used in the early 1990s, seems to be justified only in unusual cases if the diagnosis of true craniosynostosis cannot be excluded otherwise [19,34]. In our cohort of patients with distinct clinical features of positional plagiocephaly, the permeability of the sutures was detected safely and confirmed in all cases, using the 2-point method previously described [29]. Nowadays, ultrasonography is used as the first-line imaging for the diagnosis of positional plagiocephaly; however, little is known about the predictive value of sonographic skull parameters in terms of the timing and efficacy of cranial remodeling therapies, although the remodeling effect has also been shown to be easily monitorable by assessing the change of the occipital angle over time [30,35,36]. In contrast, sonographic bone thickness assessment is not a usual practice in the assessment and follow-up of positional head deformities. However, data in the literature support the validity of the technique, and not only have the differences in skull thickness related to age and sex been described, but efforts have already been made towards the creation of normative databases of age-specific data of the developing skull [37,38,39]. Moreover, correlative studies investigating the relationship between cranial bone structure and surgical outcome in craniosynostosis have been performed, although these were mostly based on computed tomography (CT) data [40,41,42,43]. From a clinical point of view, caliper cephalometry values are fundamental tools in both the classification and follow-up of positional plagiocephaly [44]. While 3D laser analysis represents an already established assessment technique, efforts towards machine learning-based monitoring tools are also being made [27,45,46]. Although orthotic helmet therapy has been widely accepted in the treatment of severe forms of positional plagiocephaly, treatment regimens remain controversial and age seems to play a role in the outcome prediction, with increasing data supporting an onset of treatment below 6 months of age. Despite the fact that there is an overall lack of Class I evidence concerning the use of molding helmet therapy in positional plagiocephaly, cranial orthoses are routinely and effectively used to treat persistent severe deformational cases [2,33]. Although available studies might mostly target patients that have obtained insufficient results from counter-positioning techniques, osteopathy and physiotherapy, the current general consensus is that cranial orthoses are at least an as efficacious, if not more efficacious, treatment, compared with more conservative approaches [2,20,21,23,47]. As a consequence, the evidence of outcome studies suggests that the more common regimen of starting helmet therapy after physiotherapy should be replaced by a combined therapy, especially in severe cases [25]. Additionally, because of the faster rate of correction in infants using helmet therapies earlier, compared with late treatment and observation, there might exist a rising pressure on healthcare providers to treat also milder forms in order to prevent long-term cosmetic issues [23]. Nevertheless, the accessibility and the treatment length of both the orthotic helmet therapy and physiotherapy can be limited by the costs, as insurance coverage is variable, and conflicts with health insurance companies regarding the refund of costs are frequently reported [28]. According to the American Association of Physical Therapy, patients fitted with cranial orthoses should receive follow-up visits one week after fitting and every two weeks thereafter, leading to costs of the cranial orthosis ranging from $1500–$3000 per orthosis, with cases of significant head growth potentially requiring a second and sometimes third orthotic helmet [28]. In addition, a standardized physiotherapy treatment plan (16 weekly 40 min physical therapy sessions) entails a cost of approximately $1200. In our cohort in France, orthotic helmet therapy is not refunded by public insurance and costs might be only partially covered by an additional private insurance, which led to an initial 15.5% drop out rate (n = 9 patients) in the recruitment of our consecutive cohort of children with deformational placiocephaly, for which a cranial orthosis was recommended. With regard to cost justification, it should be noted that, beyond the cosmetic aspects, recent data support a beneficial association between positional plagiocephaly and development in terms of cognition, although data on long-term outcomes are still scarce [48]. Nevertheless, it seems that head shape is mostly retained after the discontinuation of helmet therapy, suggesting that early efficient remodeling might be sufficient, which could also help in limiting the costs of the treatment [20,47]. Having considered all the above circumstances, we aimed at identifying sonographic parameters that could serve as a guideline for the orthotic helmet treatment of positional plagiocephaly through a fast, easily accessible, and safe predictive diagnostic tool. Our data confirmed the findings that sonographically determined occipital angles, determined as the angle between the lines projected along the lambdoid sutures of the skull, are greater than those of the unaffected side and that the OAR is positively correlated with the CVAI, as shown previously [30]. In addition, our results not only confirm the correlation between sonographically assessed changes in the occipital angle and caliper cephalometry values but provide evidence of a clear inverse correlation between both the cortical and full bone thickness of the skull and the early efficacy of an orthotic helmet remodeling of the head. Although our results confirmed that orthotic remodeling of the head is more effective at a younger age at the onset of treatment, there was no strict correlation between bone thickness and age. However, as per intention to treat, the age interval of our cohort at the onset of treatment was narrow, with the majority being four to six months old at the onset of treatment. Based on our findings, sonographically assessed cortical bone thickness and full bone thickness alone could act as predictive factors for treatment efficacy, regardless of age. However, to answer this specific question, a subsequent study with both an increased patient cohort number and a broader age range up to 12 months of age at the onset of treatment is needed to reach an adequate statistical power. Additionally, our actual study reflects that the effective time of treatment differs from the theoretic time, suggesting that also shorter treatments, as usually reported, might be effective in an early stage of the deformity. However, a subgroup analysis of the time needed to achieve correction in relation to the initial Argenta Grade and the subsequent correlation of the findings with bone thickness would be needed in the future, as available data support that the time required for deformity correction is progressively longer, according to the clinical severity at the onset of treatment [21]. Since we did include a minority of mild plagiocephaly cases in our cohort (22% Argenta Type I–II), these data do likely explain the lack of early clinical treatment efficacy in 18% of our cohort (Figure 4). Additionally, our actual study reflects that the effective time of treatment differs from the theoretic time, suggesting that also shorter treatments, as usually reported, might be effective in an early stage of the deformity. The actual difference in the treatment time is likely to be explained by the need for multiple adjustments, by side effects and by the acceptance rate of the orthotic helmet, and highlights that external cranial remodeling represents a non-invasive but incisive treatment option for positional head deformity, thus demanding targeted indications.

The findings could not only relativize the age recommendation by also allowing older infants that present with low bone thickness to profit from helmet therapy, but also lead to the targeted use of time and financial resources in patient-specific care. Moreover, our data suggest that the effective time of treatment needed to achieve objective changes in head shape might be shorter, as expected, with a reduction in the Argenta Grade by I–II points seen over 6–10 weeks in most of our cohort. 

### Limitations

Our results confirm the primary hypothesis by clearly indicating an inverse correlation between skull bone thickness, both for outer cortical and full bone thickness, providing an easily objectivable, sonographic predictive parameter for early outcome assessment in orthotic helmet remodeling of moderate-to-severe forms of posterior plagiocephaly in younger infants. However, the study design lacks a long-term results analysis. The latter, designed at least as a prospective, consecutive cohort study, might be specifically helpful in further predicting the overall length of treatment, excluding children that might not profit efficiently from orthotic helmet remodeling and in generating a more accurate cost estimate prior to the onset of treatment. In addition, a larger cohort is needed to perform a subgroup analysis, taking into account the effective duration of the therapy and the age at the onset of treatment. Finally, it would allow for an adequate cost efficacy analysis and to take the role of the cost of routine ultrasound into consideration. Finally, our population included only infants <10 months of age at the time of recruitment, thus not allowing for the assessment of the predictive value of skull thickness parameters in older children, which might still profit from an orthotic head remodeling. 

## 5. Conclusions

The sonographic assessment of skull bone thickness is a fast, safe, and widely available radiologic diagnostic tool that supports decision making in the orthotic helmet treatment of young infants with positional plagiocephaly by predicting the early therapeutic cranial remodeling effect. The measurement can easily be integrated within the framework of the sonographic diagnostic examination of suture patency with minimal outlay and might allow for the cost-effective, patient-specific indication of cranial orthosis treatment. Further prospective studies with an extension of both the follow-up period and the age inclusion are needed to assess the predictive value on the long-term outcome of orthotic helmet treatment and to allow for age correlation, potentially leading to the development of an individualized treatment planning. 

## Figures and Tables

**Figure 1 diagnostics-14-01407-f001:**
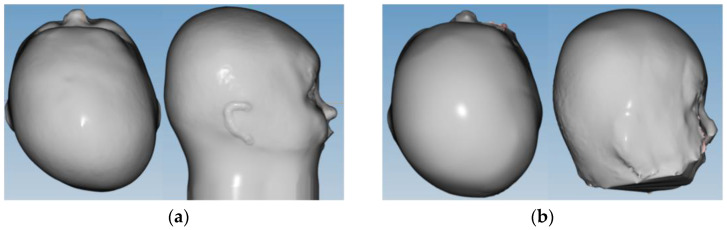
Example of 3D laser reconstruction model of positional posterior left plagiocephaly at the (**a**) onset of treatment and (**b**) early follow-up time point (3 months).

**Figure 2 diagnostics-14-01407-f002:**
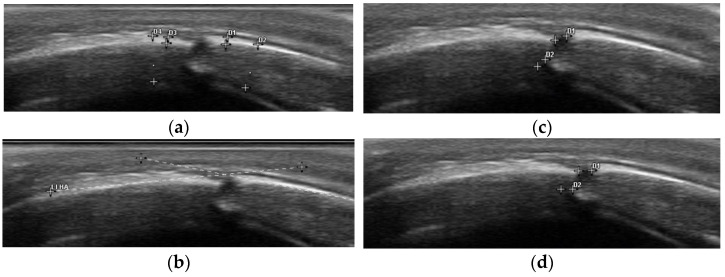
Sonographic parameter assessment adjacent to the permeable lambdoid suture (**a**) cortical and full bone thickness, assessed on both sides with extraction of the mean value; (**b**) occipital angle; (**c**) vertical suture width measured twice with extraction of the mean value; and (**d**) horizontal suture width measured twice with extraction of the mean value.

**Figure 3 diagnostics-14-01407-f003:**
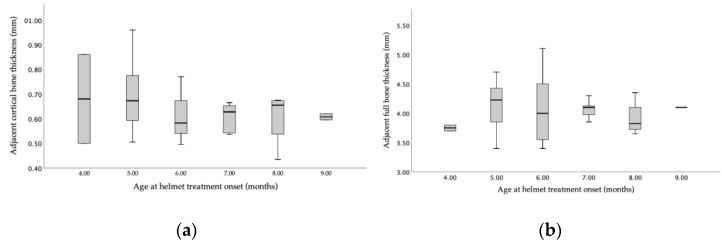
Representation of the age-related distribution of (**a**) upper cortical bone thickness (in millimeters) and (**b**) full bi-cortical bone thickness (in millimeters) at the onset of orthotic helmet treatment (n = 49 patients).

**Figure 4 diagnostics-14-01407-f004:**
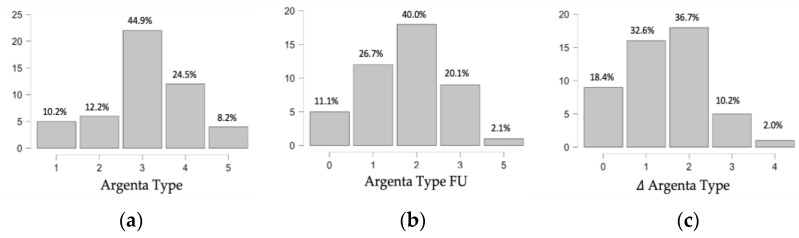
Evolution of Argenta Grading Type of positional plagiocephaly from the (**a**) onset of orthotic helmet therapy to the (**b**) early follow-up measurement after 10–14 weeks. (**c**) Change in Argenta Grading Type over time (n = 49 patients).

**Figure 5 diagnostics-14-01407-f005:**
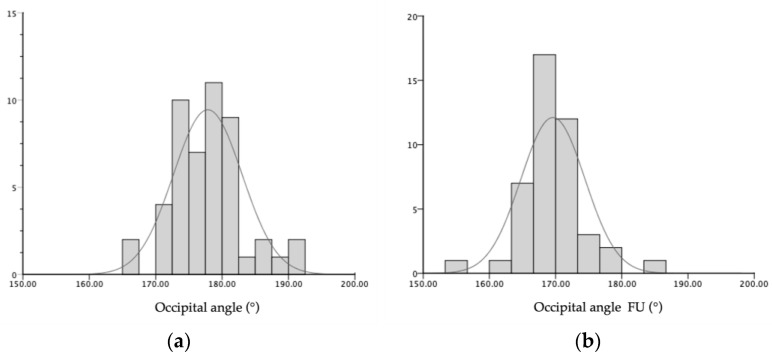
Distribution of occipital angle (°) of the affected side (**a**) at the time of recruitment and (**b**) at 10–14 weeks after the initiation of orthotic helmet therapy for the cranial remodeling of positional plagiocephaly (n = 49 infants).

**Figure 6 diagnostics-14-01407-f006:**
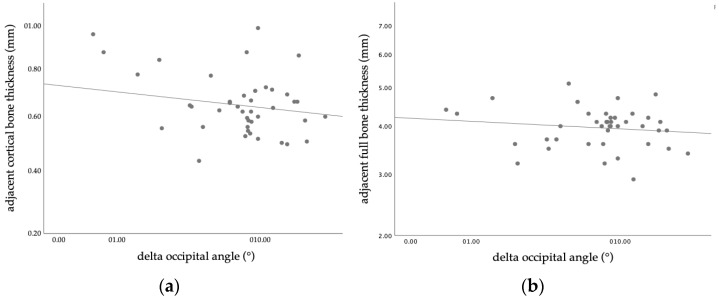
Scatterplot of inverse correlation between bone thickness and the reduction in the occipital angle of the flattened cranial bone at the three-month follow-up appointment: (**a**) upper cortical skull bone thickness (*p* = 0.033) and (**b**) full bi-cortical skull bone thickness (*p* = 0.004).

**Figure 7 diagnostics-14-01407-f007:**
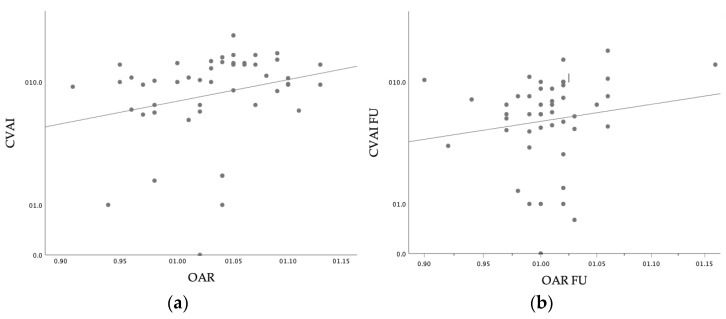
Scatterplot of linear correlation between the cephalometric cranial vault asymmetry index (CVAI shown in %) and the occipital asymmetry ratio (OAR) (**a**) at the onset of orthotic helmet therapy (*p* = 0.001) and (**b**) at a follow-up time point at 10–14 weeks (*p* = 0.042).

**Table 1 diagnostics-14-01407-t001:** Demographic characteristics, risk factors and adjuvant therapeutic maneuvers of the study cohort (*n* = 49 infants).

	nr	%
Male	34	69.4
Female	15	30.6
Prematurity ^1^	5	10.2
Gemini	4	8.2
Contralateral torticollis	30	61.2
Infant sitting	12	24.5
Tummy time >30 min/day	34	69.4
Positional sleep maneuvers	18	36.7
Osteopathy alone	7	14.3
Physiotherapy alone	8	16.3
Osteopathy + Physiotherapy	31	63.3

^1^ < 37 weeks of gestation.

## Data Availability

The data presented in this study are available on request from the corresponding author. The data are not publicly available due to ethical restrictions.

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
