# Peer review of "Predictive Value of Sonographic Parameters on the Effects of Cranial Molding Helmet Therapy in Infants with Positional Plagiocephaly"

_diagnostics, 2024, doi:10.3390/diagnostics14131407_

Round 1

Reviewer 1 Report

Comments and Suggestions for Authors

This manuscript describes a study of ultrasound measures of skull thickness and their ability to predict response of positional plagiocephaly to orthotic helmet treatment. The methods are sound and well described, and multiple other measures are used to validate the findings about plagiocephaly progression with treatment. The statistical analysis seems appropriate to the information gathered. I believe this is overall an interesting study. I think the manuscript would be improved with a few clarifications.

- The argument is made that predicting outcome of helmet treatment could help with cost-effectiveness of treatment. I think this is hard to argue from the data, first of all because there is no data presented on costs of treatment. Second, it is difficult because according to data in Figure 4, there are about 18% of the population in this study that did not have a change in Argenta type, and it is not clear how many of these could have been predicted by ultrasound measures a priori on whether not performing helmet treatment in a few patients would make up for the cost of routine ultrasound in all the patients. Third, although the authors show a correlation between ultrasound skull measures and change in occipital angle, there is not an analysis on what cutoffs could be used to predict non-response to helmet treatment. This is not a fatal flaw to the study, because there could potentially be other reasons to predict helmet response, but it is a hard thing to argue without data to support cost efficacy.

- It would help the discussion to have more information about treatment failure rate overall in helmet treatment.

- Line 238: “early 1900s” appears to be a typo, as CT would not have been available then.

Author Response

1) The argument is made that predicting outcome of helmet treatment could help with cost-effectiveness of treatment. I think this is hard to argue from the data, first of all because there is no data presented on costs of treatment. Second, it is difficult because according to data in Figure 4, there are about 18% of the population in this study that did not have a change in Argenta type, and it is not clear how many of these could have been predicted by ultrasound measures a priorion whether not performing helmet treatment in a few patients would make up for the cost of routine ultrasound in all the patients. Third, although the authors show a correlation between ultrasound skull measures and change in occipital angle, there is not an analysis on what cutoffs could be used to predict non-response to helmet treatment. This is not a fatal flaw to the study, because there could potentially be other reasons to predict helmet response, but it is a hard thing to argue without data to support cost efficacy.

Authors’Response:

We would like to thank you for the accurate revision and for your valuable input.

To support the discussion with the fact that orthotic helmet treatment is expensive and might represent a limiting factor for access to and continuation of therapy, we added specific data on treatment costs and insurance coverage. (Lines 296-309). However, we agree that our data are not sufficient to argue that cost-effectiveness could be clearly improved. In fact, the limited number of our cohort does not allow for accurate multivariate analysis of the factors influencing early treatment failure, since the reported variability in treatment adherence (hours/day) and effective length of treatment could play a combined role. With regards to the cohort number, the same applies for clear cutoff definition of sonographic occipital angle changes. Therefore, further studies including both a larger patient number and long term follow-up data would be needed in order to allow for accurate cost-estimation. This consideration was added in the limitations sector and relativizes our hypothesis on potential prediction of cost-efficacy, which we still consider as valid.

With regards to the cost of routine ultrasound assessment, we cannot provide accurate data since it is a common practice in our neurosurgical department, to assess suture permeability by ultrasound, without added costs for the patient and the health insurance. Only in cases for which the patency might not be clearly given by the method described, we do refer patients to the radiology department in order to repeat the exam and provide a legally valid radiologic report, that generates costs. With regards to the assessment of predictive sonographic factors on the effects of cranial molding helmet therapy, cost efficiency of routine ultrasound in all patients compared to treatment efficacy can only be supported by further scientific evidence and there would be need for funding for larger study designs.

2) It would help the discussion to have more information about treatment failure rate overall in helmet treatment

Authors’Response:

We do agree that precise information about treatment failure rates would be extremely helpful, however we did not find sufficient data to support the discussion. Nevertheless, we did additionally provide further considerations on the effective time needed for correction, supported by a literature review (new reference nr. 28), in relation to severity of the deformity rather than to age. These data support that time to treatment response seems to be proportionally longer with the increase of the severity grade, so the lacking early efficacy in 18% of our cohort, defined by reduction of the Argenta grade, is likely linked to the severity grade of the cranial deformity. A long-term analysis is needed to assess delayed therapeutic responses.  

3) - Line 238: “early 1900s” appears to be a typo, as CT would not have been available then.

Authors’Response:

Thank you for recognizing the error that we did obviously miss. Indeed, it’s “1990s” and we did now correct the typo.

Reviewer 2 Report

Comments and Suggestions for Authors

The research summarizes a study that investigated the predictive value of sonographic parameters on the effects of cranial molding helmet therapy in infants with positional plagiocephaly. Positional plagiocephaly is a common deformational cranial flattening in infants, often treated with orthotic helmet therapy. The study aimed to investigate the early predictive value of sonographic parameters on the efficacy of orthotic helmet therapy, by assessing changes in skull shape and correlating the parameters with caliper cephalometry values and age. 

1. The methodology is well-detailed, and the inclusion of a consecutive pediatric cohort strengthens the study's design. However, the exclusion criteria could be expanded upon. For example, explain why infants older than 12 months were excluded and discuss any potential biases this might introduce. The description of the sonographic measurements is clear, but it would be helpful to include more information on the training and calibration of the personnel performing the ultrasound assessments to ensure consistency and reliability.

2. The discussion section effectively interprets the study's findings, linking them back to the initial hypotheses. However, it would be valuable to discuss the clinical implications of these findings in more depth. How might these predictive parameters change current clinical practice? Address potential confounding factors in greater detail. For example, consider discussing how variations in helmet wear time and adherence might influence the outcomes. The study's limitations are acknowledged, but a more detailed discussion on how these limitations could be addressed in future research would be beneficial. Additionally, suggesting specific prospective study designs that could validate these findings would add value.

3. The ethical considerations are adequately addressed. It might be beneficial to mention any specific measures taken to ensure the comfort and safety of the infants during sonographic assessments. It might be beneficial to elaborate on any measures taken to minimize discomfort for the infant participants during the ultrasound procedures.

4. The limitations section is well-articulated, but consider adding more details on how future studies could address these limitations. Suggest specific research designs or additional parameters that could be explored in subsequent studies.

5. The manuscript is well-written, but there are a few minor grammatical errors and typos that need correction. 

Comments on the Quality of English Language

The manuscript is well-written, but there are a few minor grammatical errors and typos that need correction. 

Author Response

  1. The methodology is well-detailed, and the inclusion of a consecutive pediatric cohort strengthens the study's design. However, the exclusion criteria could be expanded upon. For example, explain why infants older than 12 months were excluded and discuss any potential biases this might introduce. The description of the sonographic measurements is clear, but it would be helpful to include more information on the training and calibration of the personnel performing the ultrasound assessments to ensure consistency and reliability.

Authors Response:

We thank you for the meticulous revision of our manuscript and we sincerely appreciate your valuable and precisely structured input.

Children over the age of 12 months were excluded because it has been shown that treatment efficacy is inversely related to age at onset of treatment. Data on the effect of cranial molding therapy in children over 12 months old are scarce and the results still debatable. Since we did aim at analyzing predictors for early efficacy of the cranial molding treatment, we set the age cutoff at 12 months. In fact, there were no children older than 10 months (at onset of treatment) presenting to our department for deformational plagiocephaly. These considerations have been partially made in the introduction and discussion. Furthermore, in response to your valuable comment, we did adapt our methods section, explaining the exclusion age cutoff.

All measurements were performed according to previously described techniques(2-point method) by a single senior investigator (first author). We did add this specification to the methods section. Using playing distraction or feeding, the discomfort for the infants was none to minimal.

  1. The discussion section effectively interprets the study's findings, linking them back to the initial hypotheses. However, it would be valuable to discuss the clinical implications of these findings in more depth. How might these predictive parameters change current clinical practice? Address potential confounding factors in greater detail. For example, consider discussing how variations in helmet wear time and adherence might influence the outcomes. The study's limitations are acknowledged, but a more detailed discussion on how these limitations could be addressed in future research would be beneficial. Additionally, suggesting specific prospective study designs that could validate these findings would add value.

Authors Response:

Both the Discussion and the Limitation Section were further extended, as suggested by both reviewers,  adding specific considerations on the potential effect on cost estimation and reduction. Furthermore, propositions for prospective studies investigating long-term results and including subgroup analysis in order to address the limitations of the current results were extended.

  1. The ethical considerations are adequately addressed. It might be beneficial to mention any specific measures taken to ensure the comfort and safety of the infants during sonographic assessments. It might be beneficial to elaborate on any measures taken to minimize discomfort for the infant participants during the ultrasound procedures.

Authors Response:

Using playing distraction or feeding as well as preferential positioning, according to the age, the discomfort for the infants was none to minimal. The methods section was extended accordingly.

  1. The limitations section is well-articulated, but consider adding more details on how future studies could address these limitations. Suggest specific research designs or additional parameters that could be explored in subsequent studies.

Authors Response:

Further considerations, as mentioned in response to Comment 3 and to reviewer’s 1 comments, were added in the discussion and methods section.

  1. The manuscript is well-written, but there are a few minor grammatical errors and typos that need correction.

Authors Response:

Thank you for pointing out the presence of residual minor grammatical errors and typos, that have been corrected in the actualized manuscript version.